# Revisiting the Self-Confidence and Sport Performance Relationship: A Systematic Review with Meta-Analysis

**DOI:** 10.3390/ijerph19116381

**Published:** 2022-05-24

**Authors:** Marc Lochbaum, Mackenzie Sherburn, Cassandra Sisneros, Sydney Cooper, Andrew M. Lane, Peter C. Terry

**Affiliations:** 1Education Academy, Vytautas Magnus University, 44248 Kaunas, Lithuania; 2Department of Kinesiology and Sport Management, Texas Tech University, Lubbock, TX 79409, USA; mackenzie.sherburn@ttu.edu; 3Honors College, Texas Tech University, Lubbock, TX 79409, USA; cassandra.sisneros@ttu.edu (C.S.); sydneyco@ttu.edu (S.C.); 4Faculty of Education, Health and Well-Being, University of Wolverhampton, Walsall WV1 1LY, UK; a.m.lane2@wlv.ac.uk; 5Division of Research & Innovation, University of Southern Queensland, Toowoomba, QLD 4350, Australia; peter.terry@usq.edu.au

**Keywords:** CSAI-2, competitive sport, state confidence, trait confidence, quantitative review

## Abstract

Self-confidence is a common research topic, and most applied textbooks include interventions designed to enhance athlete confidence. Our purpose was to quantify the self-confidence and sport performance literature using meta-analytic techniques. We also examined potential risk of bias indicators, and the moderation effects of study quality, sport characteristics, timing of confidence measurement, and individual differences among participants. Following a review of two past meta-analyses, a systematic search of APA PsycArticles, ERIC, Psychology and Behavioral Sciences Collection, PsychINFO, and SPORTDiscus within the EBSCOhost platform, and some hand searching, 41 articles published between 1986 and 2020 met the inclusion criteria. Collectively, the included studies investigated 3711 athletes from 15 countries across 24 sports. The overall random effects estimate of the relationship (expressed as *r*) between self-confidence and performance was 0.25 (95% CI 0.19, 0.30), with little evidence of publication bias. The summed total risk of the individual study bias score did not moderate the confidence–performance relationship, whereas significant moderator effects emerged for individual sports (0.29) compared with team sports (0.14), objective (0.29) compared to subjective (0.14) performance measures, and 100% male (0.35) compared to 100% female (0.07) samples. In conclusion, the confidence–performance relationship is small in magnitude, nearly free of bias, and moderated by sport type, performance objectivity, and athlete sex.

## 1. Introduction

Strategies to enhance self-confidence are common sport psychology interventions for athletes [1,2] but evidence of the relationship between self-confidence and athletic performance is equivocal. Several studies have reported significant benefits of self-confidence for athletes [3,4,5,6], whereas other investigations have shown no benefit [7,8,9,10]. The most recent quantitative summaries of the self-confidence in sport literature were published nearly 20 years ago in the form of two meta-analyses [11,12]. Since then, many studies are new to the literature, suggesting that an updated meta-analysis is timely. 

Self-confidence has intuitive appeal as a contributor to successful sport performance and therefore sport psychology researchers [11,12] have frequently investigated the confidence–performance relationship. Several related but distinct terms have been used in this area of the literature, including self-confidence, self-efficacy, sport confidence, or simply, confidence. For the purposes of this review, self-confidence is defined as “the perceived ability to accomplish a certain level of performance” [13] (p. 279), whereas self-efficacy is more situationally specific, and defined in sport as “a performer’s belief that he or she can execute a behavior required to produce a certain outcome successfully” [14] (p. 314). We delimited our review to studies that have explicitly evaluated the relationship between state self-confidence and sport performance, that is, used the term confidence rather than efficacy and as such excludes data sets related to self-efficacy and sport performance. We acknowledge that it is possible that researchers used the term confidence when assessing efficacy and vice versa. We use the terms sport confidence and confidence as synonyms for self-confidence as reported by athletes.

Self-confidence is both a personality trait (i.e., “a relatively stable predisposition” [14] (p. 368)) and a psychological state (i.e., “a transitory emotional condition” [14] (p. 339)). This means that some athletes, by nature, will tend to be more confident than others. However, even the most naturally confident athlete may experience low self-confidence in specific circumstances (e.g., following an unexpected defeat) or unfamiliar territory (e.g., the football quarterback on the 10 m diving board). Our review is specific to studies that evaluated the relationship between state self-confidence and sport performance and excludes investigations of trait self-confidence traits.

Previous efforts to summarize the evidence base for the benefits of self-confidence on athletic performance have included two published meta-analyses [11,12]. Woodman and Hardy [12] summarized 47 studies in their meta-analysis, 40 of which (85.1%) had used the Competitive State Anxiety Inventory-2 (CSAI-2) [15] to assess self-confidence. The mean effect size reported for the relationship between self-confidence and performance (*r* = 0.24, *p* < 0.001) represents a small positive effect [16], accounting for 5.8% of shared variance. The confidence–performance relationship was moderated by participant sex (male > female), standard of competition (high standard > low standard), and measurement scale (CSAI-2 < other measures). Craft et al. [11] limited their meta-analysis to studies using the CSAI-2 to assess self-confidence, identifying 29 studies that met their inclusion criteria. They reported an identical relationship with performance (*r* = 0.24, *p* < 0.001) to Woodman and Hardy [12], which was moderated by type of sport (individual > team), type of skill (open > closed), level of athlete (higher ability > lower ability), and time of CSAI-2 administration (31–59 min. pre-competition > other periods).

Meta-analyses related to self-confidence in domains other than sport have shown similar results. For example, a meta-analytic review of 114 studies investigating the relationship between self-efficacy and work-related performance [17] reported a weighted mean correlation of 0.38, with task complexity (low > medium > high) and study setting (laboratory > field) moderating the relationship. Further, research has demonstrated that self-efficacy is one of the strongest correlates of academic performance [18,19].

There are compelling reasons to hypothesize that characteristics of the sport in question and the athletes involved moderate the relationship between self-confidence and sport performance. In 1995, Terry [20] identified a range of factors that he proposed would moderate the relationship between mood states and sport performance, and indicated that the same moderation effects would apply to other psychological states, including self-confidence. For example, he proposed that the relationship would be stronger in individual sports than team sports because individual sports remove the influence of team dynamics. A test of this hypothesis among 100 tennis players in singles and doubles competition showed the confidence–performance relationship to be of moderate strength in singles and close to zero in doubles [21]. Further exploration of the confidence–performance relationship in team sports compared to individual sports will form part of our review. 

A related proposition [20] refers to the nature of the skills involved in the sport performance. The confidence–performance relationship would tend to be stronger in sports that involve primarily closed skills (i.e., skills performed in a predictable, unchanging environment, such as archery [22]) than in sports that involve primarily open skills (i.e., skills performed in a dynamic, rapidly changing environment, such as soccer [22]) because the direct influence of opponents is removed. However, of note, Craft et al. reported the opposite, that the confidence–performance relationship was stronger in open-skill sports than closed-skill sports [11]. Further, given the potential for self-confidence to increase or decrease after performance is underway, Terry [20] proposed that the confidence–performance relationship should be stronger in sports of shorter duration, in which performance outcome is determined closer in time to the pre-performance assessment of self-confidence. 

Another proposition relates to whether the sport performance in question is self-referenced (e.g., running a personal record time) or other-referenced (e.g., finish position). Terry [20] proposed that the confidence–performance relationship would be stronger when performance is self-referenced than when it is other-referenced. For example, running a sub-3 h marathon is an exceptional performance for an average runner even though they would have placed outside the top-1500 finishers in the 2019 New York City Marathon [23], whereas an elite runner posting a time of 2 h 30 min would have underperformed, despite finishing inside the top-70 runners in the same race. This principle extended, involves comparison of the confidence–performance relationship according to whether the performance measure is objective (e.g., win–loss, time to complete) or subjective (e.g., self-rated, coach-rated). Objective measures tend to be more precise indicators of performance than subjective measures, and therefore we expect to show a stronger relationship with self-confidence. Conversely, subjective ratings may be more sensitive to individual variations in performance. On balance, although both objective and subjective performance measures can be self-referenced (e.g., personal record time, self-rating of performance) or other-referenced (e.g., finish position, coach rating of performance), we anticipate that objective performance measures would show the stronger relationship with self-confidence. 

The next proposed moderator of the confidence–performance relationship is the relative skill level of the athletes involved [20]. For example, tennis legend Raphael Nadal will beat any club-level player in the world regardless of confidence level simply because his skill level is so much higher. Psychological factors, including self-confidence, tend to play a more important role in determining performance outcome among athletes who are homogeneous rather than heterogeneous in terms of skill and fitness. This is more likely to occur at the higher echelons of sport where athletes win or lose medals by tiny margins. Previous meta-analyses have reported a moderating effect of athlete level of competition, with stronger confidence–performance relationships found among higher-level athletes [11,12]. 

Finally, athlete sex may moderate the confidence–performance relationship, given the tendency for male athletes to report higher self-confidence than female athletes [24]. Woodman and Hardy [12] reported a stronger confidence–performance relationship for male athletes whereas Craft et al. [11] did not test the moderating effect of athlete sex. We will explore the veracity of the above propositions in our review, as far as the reported characteristics of the primary studies allow. 

### Research Questions

Given the continued interest in sport confidence, we aimed to update and extend the work of Craft et al. [11] and Woodman and Hardy [12] by aligning our inclusion criteria and pre-planned analyses to answer the following replication and extension research questions.

Q1: What is the overall relationship between a measure of state self-confidence and sport performance? Moreover, does the risk of individual study bias or across study bias (i.e., publication bias) moderate this relationship?Q2: Do Terry’s [20] sport propositions moderate the confidence–performance relationship?Q3: Does the objectivity and reference of the performance measure moderate the confidence–performance relationship?Q4: Does the time of the self-confidence measure prior to performance moderate the confidence–performance relationship?Q5: Do selected individual difference variables, namely sex and athlete sport level, moderate the confidence–performance relationship?

## 2. Materials and Methods

This systematic review with meta-analyses reported followed Page et al.’s [25] Preferred Reporting Items for Systematic Reviews and Meta-Analysis (PRISMA) publication. Although we did not register our review protocol with the PROSPERO database, we did search the database before and after conducting our systematic review to check that we were not duplicating a recent study. 

### 2.1. Eligibility Criteria and Selection Process

Included studies, in any printed language, met the following criteria: (a) use of a state self-confidence measure; (b) measure of sport performance; (b) self-confidence assessed before sport performance; (c) data provided, group differences (e.g., winners, losers) or relational (e.g., correlation between self-confidence and performance) for effect size calculation; and (d) original data published in peer-reviewed scholarly journals by 1 December 2021. We did not consider participants in a non-athletic setting, such as Gould and colleagues’ [26] police academy participants. We discriminated self-confidence from self-efficacy by the questionnaire used and thus the language used to describe the construct. For sport performance, we excluded physical performance measures associated with athletic performance, such as vertical jump height, but included measures of performance skills (e.g., tennis serving percentage). All authors reviewed articles considering our inclusion criteria. We did not inquire about missing data or clarifications. We imposed no language restriction. Articles found in our search resulted from the search terms found in either the title, keywords, or abstract. If an article itself was not in English (*n* = 2), then all pertinent non-English articles sections were copy and pasted into Google Translate (https://translate.google.com/, accessed on 1 March 2021).

### 2.2. Information Sources, Search Strategy, and Search Protocol

The systematic search included references from the two published state confidence and performance meta-analyses [11,12] and databases found within EBSCOhost (search ended 1 December 2021). The specific databases were APA PsycArticles, ERIC, Psychology and Behavioral Sciences Collection, PsychINFO, and SPORTDiscus. The search terms were combinations of sport performance with confidence, competitive, CSAI-2, TSCI, state confidence, and sport confidence measures. Our Appendix A contains the record of our complete protocol. When reading the Appendix A, the term ‘Box’ refers to the EBSCO interface, whereby one types search terms in the advanced search setting. The following is an example of our search protocol:EBSCOAPA PsycArticles, APA PsychINFOERIC, Psychology and Behavioral Sciences Collection, SPORTDiscusTyped key termsCSAI-2 typed in Box 1 in the EBSCO interfaceSport performance typed in Box 2 in the EBSCO interfaceCompetitive typed in Box 3 in the EBSCO interfaceLimit to scholarly peer-reviewed journalsPage options: 50Limit time: 2002–2020 [computer changed to 2003]

### 2.3. Study Selection

We checked the title, abstract, and keywords of all articles identified in our search procedure for mention of the term confidence, but not confidence interval(s). Articles mentioning confidence went forward for full-text assessment. If there was any doubt about whether a study included confidence as a measured variable despite having no mention of it in the title, abstract, and keywords, we retained the study for full-text screening. ML, MS, and CS assessed studies for eligibility, independently and then collectively. AL and PT then independently verified that all included studies met the inclusion criteria and identified two additional eligible studies. Figure 1 depicts our search process via a PRISMA 2020 flow diagram template (http://prisma-statement.org/prismastatement/flowdiagram.aspx, accessed on 1 March 2021).

### 2.4. Data Collection and Items Retrieved

ML and MS planned the extraction of data. After examining the two past meta-analyses [11,12], they examined Beedie et al. [27] and Lochbaum et al. [28], given that the methodologies of mood states measured before sport performance is akin to confidence measured before sport performance. ML and MS, independent of each other, began the data extraction. Next, all remaining authors received data collection training. ML worked with each to finalize the retrieval process. The data items retrieved were as follows: sample age, country, number, percent male, and sport; confidence (name of scale) and performance specifics (objective or subjective, other- or self-referenced); and the sport type (individual or team), duration (less than or greater than 10 min in duration), and skill (closed or open).

### 2.5. Risk of Bias in Individual Studies

We used Hoy and colleagues’ [29] risk of bias tool to rate risk of individual study bias. At the outset, ML, CS, MS, and SC worked on the coding for the potential of individual study risk of bias in groups. Next, AL, PT, and ML worked to consensus on each risk category. We coded all studies (see Table 1) on the following nine risks of bias: (1) target population is a close representation of the national population in relation to relevant variables; (2) some form of random selection was used to select the sample; (3) likelihood of nonresponse bias is minimal; (4) data were collected directly from the participants; (5) confidence measure has validity and reliability; (6) performance measure is valid and relevant to sport; (7) same mode of data collection was used for all participants; (8) assurance of participant anonymity is stated in methods; (9) assessment period for the parameter of interest is appropriate. We assigned point values to each of the three ratings (high, medium, and low), and computed a study quality score (range 9 to 27 points), where higher scores equals lower risk.

### 2.6. Summary Measures and Planned Methods of Analysis

We chose the correlation coefficient (*r*) as the primary effect size parameter. We followed Cohen’s [30] interpretation for correlation values of 0.10–0.29 as small, 0.30–0.49 as medium, and 0.50 or greater as large. We assumed heterogeneity of effects. Thus, we planned both random- and mixed-effects analyses. We based our expectancy of heterogeneity on a recent systematic review of meta-analyses in sport psychology with performance as an outcome [31]. For the test of the state self-confidence and performance relationship, we used a random-effects model. We reported the number of cases, sample size, *r*, 95% confidence intervals, heterogeneity, and publication bias statistics. We looked at heterogeneity measured as inconsistency and reported the *I*^2^ statistic or the ratio of excess dispersion to total dispersion. Higgins et al. [32] state that *I*^2^ is the overlap of confidence, explaining the total variance attributed to the covariates. To help interpret *I*^2^, Higgins and Thompson [33] suggested a tentative classification of 25 (low), 50 (medium), and 75 (high). For our moderator tests, study quality, Terry’s [20] propositions, performance measure characteristics, and athlete standard, we used a mixed-effects analysis. For these analyses, we reported the number of cases, sample size, *r*, 95% confidence intervals, and the Q total between (Q_TB_) with associated *p*-value. The Q_TB_ indicates the level of difference between different moderator levels. We used a random-effects meta-regression model to test the impact of percent male athletes on the confidence–performance relationship. Last, we examined our results with the aim of assessing certainty. We conducted our meta-analyses using the Comprehensive Meta-Analysis (CMA) version-3 software (version 3.3.070, Biostat, Inc., Englewood, NJ, USA, 20 November 2014).

### 2.7. Risk of Bias across Studies

We examined publication bias as the risk of bias across studies. We examined the fail-safe *n* calculation, the funnel plot, and the ‘trim and fill’ results for random effects found in the CMA program. The fail-safe *n* statistic is the number of null effect samples required to change a significant effect size into a non-significant effect size [34]. The greater the value, the greater the confidence that the meta-analyzed result is indeed safe from publication bias (based on the one-tail test in our analyses). Thus, the larger number of studies per reported study value, the greater the confidence in the effect size being free of publication bias. The random-effects funnel plot of precision determined whether the entered studies dispersed equally on either side of the overall effect [35], as symmetry theoretically represents the notion that entered studies captured the essence of all relevant studies. Concerning sample size and the funnel plot, smaller studies are closer to the bottom and larger studies closer to the top of the graph. We used Duval and Tweedie’s [36] trim and fill analysis to fix asymmetry. The data points are either filled to the left (i.e., lowering the effect size value) or right (i.e., increasing the effect size value) of the mean.

## 3. Results

### 3.1. Study Selection and Characteristics

Table 2 shows the 41 studies meeting all inclusion criteria of which a few provided more than one data set for 49 total samples. None of the studies provided data from more than one sport nor more than one country. The studies spanned from 1986 to 2020, with 3711 total participants (range = 7–416, mean = 80.84, SD = 92.13) with data coming from 15 countries on the following continents: Australia and Oceania, Europe, and North and South America. Most participants were adolescents and young adults, as only three sample age group means were greater than 30 years of age. Of the studies identifying participant sex, the majority were males with a mean percent male of 65.46 (SD = 36.25) and a range from 0% to 100%. The included studies contained a variety of sports, both individual (e.g., boxing, golf, taekwondo) and team (e.g., basketball, softball, volleyball).

### 3.2. Risk of Bias within Studies

The Appendix A contains all the risk of bias within studies details. The two samples from Bejek and Hagtvet [8] differed in methodology; hence, both samples are listed. Across the risk of bias topics, the mean score (possible range 9–27 points) was 20.58 (SD = 1.76). We rated each study as either high (*n* = 15), medium (*n* = 9), and low (*n* = 19) quality (i.e., risk of bias) based on being above, below or at the median score. The median score of 21 is the medium quality group. We tested study quality as a moderator of the overall confidence–performance relationship. The mixed effects Q_Tb_ statistics was not significant (*p* = 0.37), and the random effects *r* [95% CI] were as follows for the three quality of rating groupings: high 0.27 [0.16, 0.37], medium 0.29 [0.18, 0.40], and low 0.20 [0.11, 0.29]. Each random effects mean *r* was different from zero (*p* values < 0.001).

### 3.3. Results of Individual Studies, Synthesis of Results, and Risk of Bias across Studies

Figure 2 contains the individual study data. From those data, the overall effect of the confidence–performance relationship was different from zero (Z-value = 8.30, *p* < 0.001), small in magnitude with a point estimate calculated as *r* of 0.25, and with medium-to-high heterogeneity (*I*^2^ = 64.49). The 95% confidence intervals spanned from small (0.19) to medium (0.30) in magnitude. As represented in Figure 3, little publication bias existed in the data (trim *n* = 1, adjusted point estimate of 0.24% and 95% confidence intervals, ranging from 0.19 to 0.30), were almost identical to the non-trimmed results. Last, relative to the 49 samples from 41 studies, the fail-safe *n* was large at 3601.

### 3.4. Moderators

#### 3.4.1. Terry’s Propositions and Performance Characteristics

Table 3 contains our coded moderators, and Table 4 contains the mixed-effects analysis results. Concerning Terry’s [20] propositions, the confidence–performance relationship was higher for sports of less than 10 min, closed skilled, and individual when compared to sports greater than 10 min, open skilled, and team. The sport type mixed-effects analysis was significant at the traditional level (*p* < 0.05). The upper 95% confidence intervals for sports of less than 10 min, closed skilled, and individual were all medium (*r* > 0.30) in magnitude. For the two performance characteristic moderators, the effect size values were greater in magnitude for objective compared to subjective (*p* < 0.05) and other-referenced compared to self-referenced performance measures. The upper 95% confidence intervals for objective and other-referenced performance measures were medium (*r* > 0.30) in magnitude.

#### 3.4.2. Timing of Confidence Measure and Individual Difference Moderators

For the confidence measurement before performance period, all periods mean values were small in magnitude. The upper 95% confidence interval crossed the medium threshold (*r* > 0.30) for each time category. Last, for our attempts to examine individual difference moderators, neither the Craft et al. [11] nor Woodman and Hardy [12] moderator categories differed, and all mean values were small in meaningfulness. However, results for percent male participants were significant in both the meta-regression (see Figure 4) and mean difference analyses.

### 3.5. Certainty of Evidence 

Table 5 contains our research questions, our rating of certainty of evidence corresponding to the research question, and the basis for our certainty rating. The Appendix A contains details of data extracted and compared with our results and those of Craft et al. [11] and Woodman and Hardy [12].

## 4. Discussion

The present study was a systematic review with meta-analysis of the published literature on the state of the confidence–performance relationship. We distinguished self-confidence from self-efficacy based on the terms used by authors in their article. Overall, results showed that self-confidence has a positive effect on performance, moderated by sport-type, measure of performance, and athlete sex. Our findings mirrored those of the two past meta-analyses [11,12]. Given the congruencies and the minimal bias in our data, the certainty is high that the confidence–performance relationship is small in magnitude. Even at its strongest, the relationship in our mean level data rarely crossed the moderate threshold (*r* > 0.30) in meaningfulness. 

An intriguing question is why is the confidence–performance relationship not as strong as theory would predict? Michie et al. [69] proposed self-confidence impacts performance via mechanisms such as increasing effort, selecting appropriate strategies, and regulating unwanted emotions. One explanation is that the central premise of higher confidence leading to better performance is overstated. To address this question, researchers would need to test the mechanisms by which the confidence–performance relationship occurs, which relies on using methods that enable the accurate detection of the influence of self-confidence on performance. It is apparent by reviewing the 41 included studies and the literature overall, that researchers have focused primarily on testing the strength of the confidence–performance relationships and have given little attention to investigating the mechanisms that underpin them. Thus, future research should investigate possible confidence–performance mechanisms.

Concerning Terry’s [20] propositions, our results showed a stronger relationship between confidence and performance in short duration sports compared to longer duration sports and individual sports compared to team sports. These results speak to the idea that self-confidence may change during performance, either increasing or decreasing. In team sports, self-confidence could be dependent upon teammates’ actions and confidence could change once performance begins. Thus, the longer the sport event, the greater the possibility that confidence changes and thus the confidence–performance relationship weakens. Likewise, in team sports, one individual’s confidence may have little to do with performance outcome if they get minimal playing time and/or the actions of teammates determine the performance outcome. It is possible that self-confidence will be influential on performance, but to detect its effect requires a more sensitive research design, such as repeated measures within the event. Emotion research has faced similar challenges, whether it be measuring emotions prior to performance [28] or using retrospective designs [70]. In our included studies, only Totterdell [66] used a repeated-measures design. A few excellent examples in the emotion research exist whereby researchers captured multiple emotion–performance relationships during competition [71,72]. Hence, to understand the confidence–performance relationship, we suggest repeated confidence testing within the event while recording performance. 

Continuing our questioning as to why the confidence–performance relationship is small though moderated by performance characteristics (i.e., type and reference), another issue when measuring self-confidence is the extent to which participants have accurate knowledge of tasks demands. For example, if people have recently completed the task, then they have an experiential basis to rate future expectancies. Bandura [73] highlighted that when there is an abundance of feedback on a specific task, confidence estimates tend to mirror previous performance closely. Such an assertion works well when the performance task remains stable, such as with a math puzzle. In sports competition, even closed skills contain factors that change and when the level of competition rises, and differences between winners and losers are marginal, such uncontrollable factors grow in importance.

A further issue when assessing self-confidence is the extent to which people have access to relevant information on which to base self-confidence estimates. Athletes may base their confidence on belief in skill execution, physical fitness, and intended effort, but confidence estimates remain hypotheses until tested by situational factors. There is an ongoing feedback loop between behavior and perception, whereby athletes assess and re-assess their confidence estimates from continuous performance feedback. Failing to meet the standard expected tends to activate unpleasant emotions that serve as a signal to improve performance possibly by increasing effort or changing strategy. However, at the time athletes self-report their confidence, the information used comes from memories distal from the current competition. Therefore, given the importance of having accurate and available information to inform self-confidence ratings, objective (vs. subjective) and other-referenced (vs. self-referenced) performance measures are more accurate and available to the performer.

A limitation of research investigating sport confidence is that neither the information used by athletes to rate self-confidence, nor the strategies they plan to use to achieve their goals are known. Therefore, we do not know how athletes arrive at their ratings, only what ratings they provide. Research has been conducted into the antecedents of self-confidence ratings among athletes [74,75], which provides additional insights. However, further exploration, perhaps using qualitative techniques, of exactly how athletes arrive at their self-confidence ratings and develop their beliefs about the effectiveness of planned competition strategies would be particularly germane.

Examination of the evidence base and strategies to deliver performance expectations might be particularly useful when exploring sex differences in the confidence–performance relationship. Although male athletes tend to report higher self-confidence compared to female athletes [24], this does not in itself explain sex differences in the confidence–performance relationship. Using intra-individual analysis of performance in the shooting phase of 254 international biathlon competitions, Ahammer et al. [76] showed that a one standard deviation reduction in self-confidence increased the number of missed shots by 0.53 standard deviations for men, but there was no effect of self-confidence on missed shots for women. Further investigation of sex-based differences in cognitive, behavioral, and emotional processes that occur between the pre-competition assessment of self-confidence and the outcome of performance may provide valuable insights into the confidence–performance relationship. Last, an interesting observation relates to the age of samples used to study self-confidence. Nearly all samples were younger adults, with mean age exceeding 30 in only three studies. With aging and gathering experiences, athletes might accrue greater knowledge of task demands and therefore provide more accurate confidence estimates. We suggest that future research should investigate the confidence–performance relationship in targeted populations, such as all-female and master athletes. 

### Study Limitations

Even having closely followed the PRISMA statement [25], limitations within our meta-analysis are evident. First, although we identified 41 studies meeting our inclusion criteria, it is possible we missed relevant studies because the CSAI-2 measure is more closely associated with anxiety than with confidence, and article titles, abstracts, and keywords might make no mention of confidence despite having measured it. Likewise, studies with multiple psychological measures might exclude mention of confidence in the title, abstract, and keywords. We ameliorated this potential limitation by retaining studies for full-text screening if there was even a suspicion that confidence was a measured variable despite not being mentioned in the title, abstract, nor keywords. Second, we decided, given the decades covered in our search, from the 1980s or potentially earlier to the 2020s, not to contact authors for missing data. Our reasoning was the passage of time for data storage and even deceased researchers would bias the data available. However, the minimal publication bias found eased our concerns of these two limitations. Third, with 41 studies contributing 49 samples, small sample sizes were present in some of our moderator analyses. Smaller samples limited statistical power to detect significant between-level differences and may have contributed to larger confidence intervals. Fourth, we attempted to test the individual difference moderators reported in Craft et al. [11] and Woodman and Hardy [12]. Without exact operational definitions and coding for levels such as elite, European club, and top and lower standard, we may have coded our samples differently to previous research teams. Last, we sought to include eligible studies with no language restriction. To do so, we used Google Translate (https://translate.google.com/, accessed on 1 March 2021). It is possible that Google Translate is not 100% accurate and we either excluded eligible studies or included ineligible studies. Although we have mentioned use of Google Translate as a limitation, including studies without a language restriction was a clear strength of our meta-analysis, instead of including only studies published in English. 

## 5. Conclusions

Self-confidence dominates the sport media and the athletic rhetoric as vital to performance, in such statements such as “If you don’t have confidence, you will always find a way to not win” (Carl Lewis, 9-time Olympic Gold Medalist). However, based on our meta-analysis and two past meta-analyses, the confidence–performance relationship is small in magnitude with a few important moderators. It might be true, as Carl Lewis asserts, that without confidence one cannot win. However, it might be simply that without more confidence than the other team or competitor at a critical moment, one will find a way not to win.

## Figures and Tables

**Figure 1 ijerph-19-06381-f001:**
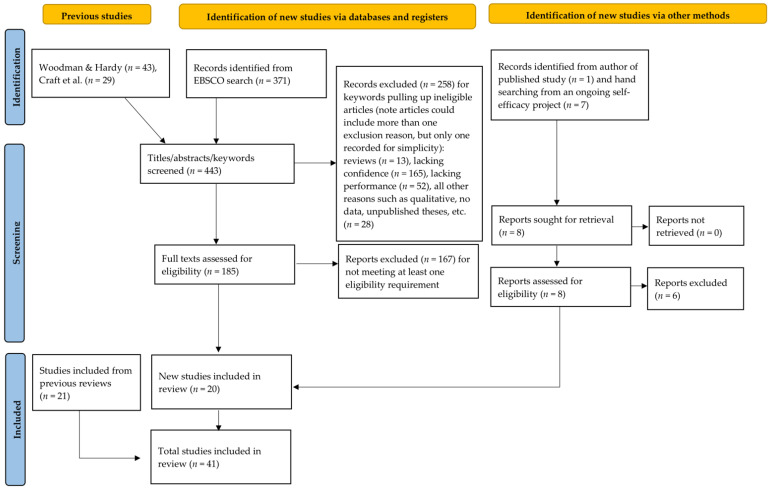
PRISMA flow chart for the identification of the included studies.

**Figure 2 ijerph-19-06381-f002:**
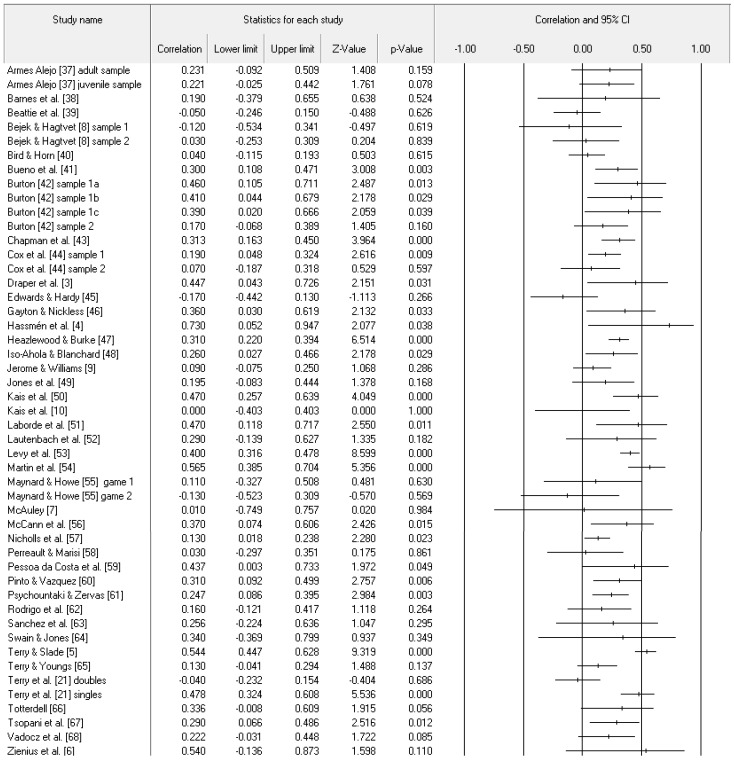
Study effect size statistics expressed as correlations and corresponding forest plot.

**Figure 3 ijerph-19-06381-f003:**
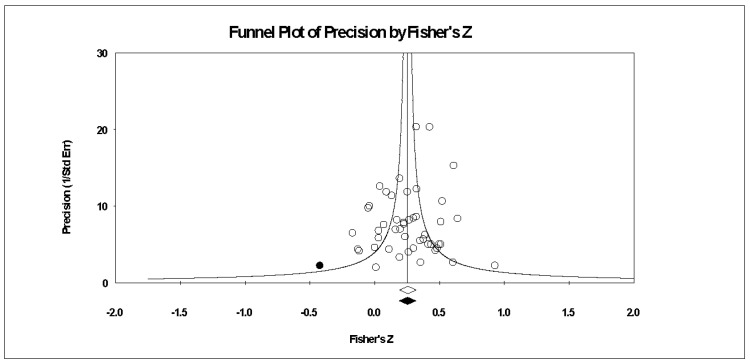
Random effects plot trimmed and filled. The open circles are the data points, and the one filled circle is the result of the trim and fill analysis. The clear rhombus is the point estimate, and the filled rhombus is the trim and filled point estimate.

**Figure 4 ijerph-19-06381-f004:**
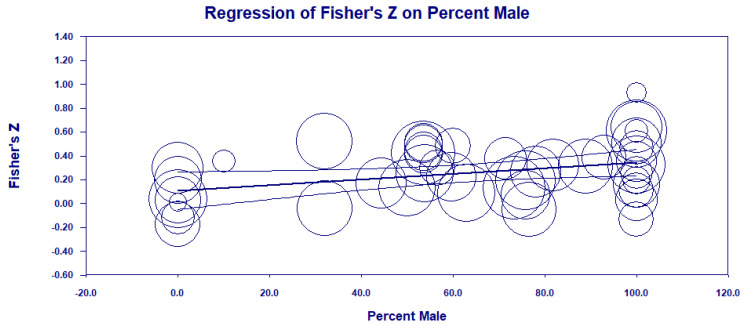
Meta-regression plot of the relationship of percent male in each study sample and the overall self-confidence and sport performance relationship, *R*^2^ = 0.22, *F*(1,43) = 6.62, *p* < 0.01, Goodness of fit Q(43) = 112.73, *p* < 0.0001, expressed as Fisher’s Z. Larger circles represent studies with more participants. Middle line is the regression line; the upper and lower lines are the 95% confidence intervals.

**Table 1 ijerph-19-06381-t001:** Risk of individual study bias questions and rating explanations.

	Rating Explanations		
Bias	Low Risk (3 Points)	Medium Risk (2 Points)	High Risk (1 Point)
Sample	Sample is a group of athletes, in the sport; performance is score, outcome of that sport.	Sample is like the high category, but sport outcome is like the low category or vice versa.	Sample includes recreational athletes, but not in a realistic sport context, and/or performance is based on assessment of a skill rather than a game statistic or sport outcome.
Random selection	Stated random selection occurred from a much larger group (e.g., from all athletes at an event).	Random selection occurred within a group of athletes (e.g., college team at an event).	No random selection of any kind stated.
Nonresponse	Appears most participants completed the measures.	Some non response occurred.	Seems most did not do it, e.g., a big race, subjects recruited at the race, most likely most did not do it.
Direct collection	Yes, all in person.	A mix of online and in person.	All online or mail.
Confidence measure	Study level reliability presented.	Valid and reliable measure (e.g., all except the 1-item measure) but study level reliability not presented.	Made up confidence measure.
Sport measure	Event time, win–loss, outcome, golf score, judge rated (gymnastics).	Participant statistics.	Self or other rated subjective, or vague (good vs. bad performers), team selection.
Data collection	Yes, all the same.	No option for a medium rating.	A mix of ways (e.g., individual for some, in large groups of others).
Anonymity	Yes, stated.	Participants presented informed consent.	Not stated in methodology.
Time	≤15 min	16–60 min	>1 h

**Table 2 ijerph-19-06381-t002:** Study characteristics.

	Participant Characteristics					Confidence and Performance Characteristics	
Study	Age (yr.)	Ctry.	N	% Male	Sport	Confidence	Performance
Armes Alejo [37]	17.00, 22.60	BR	60	NA	Boxing	CSAI-2	Medalists, non-medalists
Barnes et al. [38]	College	US	13	100	Swim	CSAI-2	Event time
Beattie et al. [39]	22.50	UK	81	76.54	Canoeing	CSAI-2	Event time
Bejek & Hagtvet [8]	Adolescent	HU	20	0	Gymnastics	CSAI-2	Event score
Bejek & Hagtvet [8]	Adolescent	HU	49	0	Gymnastics	CSAI-2	Event score
Bird & Horn [40]	14.00–17.00	US	161	0	Softball	CSAI-2	Coach rating of athlete mental errors
Bueno et al. [41]	31.01	ES	90	88.89	Endurance	CSAI-2	Successful, not successful achieving objective
Burton 1988 [42]	18.00–23.00	US	28	53.57	Swim	CSAI-2	Event performance compared to personal best
Burton 1988 [42]	17.40	US	70	44.28	Swim	CSAI-2	Event performance compared to personal best
Chapman et al. [43]	21.23	UK	142	100	Taekwondo	CSAI-2	Winners, losers
Cox et al. [44]	NR	US	248	75.81	Basketball	ARS-2	Margin of victory
Draper et al. [3]	25.60	NZ	20	60	Rock climbing	CSAI-2R	Completed route, not completed route
Edwards & Hardy [45]	21.80	UK	45	0	Netball	CSAI-2	Athlete rated performance
Gayton & Nickless [46]	34.74	US	35	71.42	Running	TSCI	Event time
Hassmén et al. [4]	21.00	SE	8	100	Golf	CSAI-2	Season score average
Heazlewood & Burke [47]	NR	AU	416	NA	Triathlon	CSAI-2	Event time
Iso-Ahola & Blanchard [48]	NR	US	73	78.08	Racquetball	1-item	Athlete standard-winners, losers
Jerome & Williams [9]	41.00	US	143	62.94	Bowling	CSAI-2	Event performance compared to league average
Jones et al. 1993 [49]	14.00–16.00	UK	48	0	Gymnastics	CSAI-2	Good performers, poor performers
Kais et al. 2004 [50]	28.20	EE	66	100	Volleyball	CSAI-2	Expert rating of athlete performance
Kais et al. 2005 [10]	NR	EE	24	NA	Team sports	CSAI-2 (Int.)	Coach rating of athlete performance
Laborde et al. [51]	23.88	DE	28	53.57	Tennis	CSAI-2	Tennis serving errors
Lautenbach et al. [52]	24.04	DE	23	56.52	Tennis	CSAI-2	Tennis serving errors
Levy et al. [53]	21.63	UK	415	53.49	NR	SSCI	Athlete rated performance satisfaction
Martin et al. [54]	16.00	US	73	100	Running	SSCI	Event time
Maynard & Howe [55]	19.00–24.00	CA	22	100	Rugby	CSAI-2	Coach rating of athlete performance
McAuley [7]	College	US	7	0	Golf	CSAI-2	Event score
McCann et al. [56]	19–26	US	42	92.86	Cycling	CSAI-2	Time maintained pace
Nicholls et al. [57]	21.30	UK	307	73.28	Mix	CSAI-2R	Athlete rated performance satisfaction
Perreault & Marisi [58]	25–40	CA	37	100	Basketball	CSAI-2	Event performance statistics
Pessoa da Costa et al. [59]	17.00	**BR**	16	100	B. volleyball	CSAI-2R	Performance statistics
Pinto & Vázquez [60]	16.14	**AR**	77	81.82	Golf	CSAI-2	Event placement
Psychountaki & Zervas [61]	11.20	GR	143	53.85	Swim	SSCQ-C	Coach rating of athlete performance
Rodrigo et al. [62]	18.00–31.00	UY	51	100	Soccer	CSAI-2	Athlete and observer rated performance
Sanchez et al. [63]	24.60	GB	19	100	Rock climbing	CSAI-2	Event score
Swain & Jones [64]	21.10	UK	10	100	Basketball	CSAI-1	Performance statistics
Terry & Slade [5]	25.35	UK	199	100	Karate	CSAI-2	Winners, losers
Terry & Youngs [65]	20.40	UK	128	50	Field hockey	CSAI-2	Team selection
Terry et al. [21]	19.90	UK	100	32	Tennis	CSAI-2	Winners, losers
Totterdell [66]	26.00	UK	33	100	Cricket	UWIST	Batting/bowling average
Tsopani et al. [67]	11.50	GR	74	0	Gymnastics	CSAI-2	Event score
Vadocz et al. [68]	15.39	UK	57	59.65	Roller skating	CSAI-2	Medalists, non-medalists
Zienius et al. [6]	16.70	LT	10	100	Golf	CSAI-2	Event score

**Bold** country abbreviation = study written in non-English. Country abbreviations: Country (Ctry.), Australia (AU), Argentina (AR), Brazil (BR), Canada (CA), Estonia (EE), Germany (DE), Greece (GR), Hungary (HU), Lithuania (LT), New Zealand (NZ), Spain (ES), Sweden (SE), Uruguay (UY), United Kingdom (UK), United States (US); Sport abbreviation: Beach (B). Confidence abbreviations: Cognitive Somatic Anxiety Inventory (CSAI), Revised (R), State Sport Confidence Questionnaire for Children (SSCQ-C), University of Wales Institute of Sciences and Technology (UWIST).

**Table 3 ijerph-19-06381-t003:** Moderator Coding for Terry’s Proposition, Performance Characteristics, and Confidence Time to Event Measured.

	Terry’s Propositions			Performance			Athlete	
Study	Time	Skill	Type	Type	Reference	Time to Event	Groupings	Standard
Armes Alejo [37]	>10	O	I	OBJ	OTH	>1 h	Elite	High
Barnes et al. [38]	<10	C	I	OBJ	Self	16–30	College	High
Beattie et al. [39]	<10	O	I	OBJ	OTH	16–30	Elite	High
Bejek & Hagtvet [8]	<10	C	I	SUB	OTH	16–30	Elite	High
Bejek & Hagtvet [8]	<10	C	I	SUB	OTH	16–30	Rec	Low
Bird & Horn [40]	>10	O	T	SUB	Self	31–60	HS	Low
Bueno et al. [41]	>10	C	I	SUB	Self	>1 h	Mix	?
Burton 1988 [42]	<10	C	I	OBJ	Self	31–60	College	High
Burton 1988 [42]	<10	C	I	OBJ	Self	31–60	College	High
Chapman et al. [43]	<10	O	I	OBJ	OTH	31–60	Rec	Low
Cox et al. [44]	>10	O	T	OBJ	Self	<15	Rec	Low
Draper et al. [3]	<10	C	I	OBJ	OTH	<15	Rec	Low
Edwards & Hardy [45]	>10	O	T	SUB	Self	<15	EC	Low
Gayton & Nickless [46]	>10	C	I	OBJ	OTH	<15	Rec	Low
Hassmén et al. [4]	>10	C	I	OBJ	OTH	31–60	Elite	High
Heazlewood & Burke [47]	>10	C	I	OBJ	OTH	>1 h	Rec	Low
Iso-Ahola & Blanchard [48]	>10	O	I	OBJ	OTH	<15	Rec	Low
Jerome & Williams [9]	>10	C	I	OBJ	Self	31–60	Rec	Low
Jones et al. 1993 [49]	<10	C	I	SUB	OTH	<15	EC	Low
Kais et al. 2004 [50]	>10	O	T	OBJ	OTH	>1 h	Elite	High
Kais et al. 2005 [10]	>10	O	T	SUB	Self	31–60	Elite	High
Laborde et al. [51]	<10	C	I	OBJ	OTH	<15	Mix	?
Lautenbach et al. [52]	<10	C	I	OBJ	OTH	<15	Rec	Low
Levy et al. [53]	Both	Both	Both	SUB	Self	31–60	Mix	?
Martin et al. [54]	<10	C	I	OBJ	OTH	16–30	HS	Low
Maynard & Howe [55]	>10	O	T	SUB	Self	31–60	College	High
McAuley [7]	>10	C	I	OBJ	OTH	<15	College	High
McCann et al. [56]	<10	C	I	OBJ	OTH	<15	Elite	High
Nicholls et al. [57]	?	?	Both	SUB	Self	>1 h	Rec	Low
Perreault & Marisi [58]	>10	O	T	OBJ	OTH	16–30	Elite	High
Pessoa da Costa et al. [59]	>10	O	T	OBJ	OTH	16–30	Elite	High
Pinto & Vázquez [60]	>10	C	I	OBJ	OTH	>1 h	USAC	Low
Psychountaki & Zervas [61]	<10	C	I	SUB	Self	31–60	EC	Low
Rodrigo et al. [62]	>10	O	T	SUB	Self	31–60	Elite	High
Sanchez et al. [63]	<10	C	I	OBJ	OTH	16–30	Elite	High
Swain & Jones [64]	>10	O	T	OBJ	OTH	16–30	EC	High
Terry & Slade [5]	<10	O	I	OBJ	OTH	31–60	Rec	Low
Terry & Youngs [65]	>10	O	T	OBJ	OTH	31–60	College	High
Terry et al. [21]	>10	C	I	OBJ	OTH	31–60	Rec	Low
Totterdell [66]	>10	O	T	Both	Self	<15	Elite	High
Tsopani et al. [67]	<10	C	I	OBJ	OTH	31–60	EC	Low
Vadocz et al. [68]	Both	Mix	Mix	OBJ	OTH	>1 h	Elite	High
Zienius et al. [6]	>10	C	I	OBJ	OTH	<15	EC	Low

Abbreviations: L = low, M = medium, H = high; ? = not enough information presented to decide, O = open skill sport, C = closed skill sport, I = individual sport, T = team sport, OBJ = objective, SUB = subjective, OTH = performance other-referenced, Self = performance self-referenced; Rec = recreational, EC = European Club, USAC = United States of America club, HS = High school; Groupings are based on our attempt to examine Craft et al. [11]; Standard is our attempt to examine Woodman and Hardy [12].

**Table 4 ijerph-19-06381-t004:** Moderator Results for Terry’s Proposition, Performance Characteristics, and Confidence Time to Event Measured.

Moderator	Group	k	*n*	r	95% CI LL	95% CI UL	Q_TB_	*p*-Value
Sport time	<10	19	1144	0.304	0.213	0.389		
	>10	27	2034	0.201	0.128	0.281	2.677	0.102
Sport skill	Closed	25	1591	0.280	0.194	0.362		
	Open	21	1587	0.212	0.122	0.298	1.202	0.273
Sport type	Individual	32	2295	0.288	0.221	0.353		
	Team	14	883	0.142	0.032	0.249	5.199	0.023
Performance type	Objective	35	2527	0.290	0.223	0.357		
	Subjective	13	1397	0.137	0.027	0.245	5.591	0.018
Performance reference	Other	30	2065	0.289	0.218	0.358		
	Self	19	1892	0.187	0.098	0.273	3.197	0.074
Confidence time	<15	13	627	0.246	0.120	0.364		
	16–30	9	336	0.187	0.024	0.340		
	31–60	19	1886	0.255	0.165	0.340		
	>1 h.	8	1108	0.274	0.146	0.393	0.771	0.856
Athlete level #	Elite	14	566	0.215	0.109	0.316		
	College	8	277	0.221	0.061	0.369		
	European club	6	330	0.195	0.038	0.343	0.062	0.969
Athlete standard ^	Higher	22	843	0.220	0.121	0.314		
	Lower	25	2609	0.252	0.177	0.324	0.270	0.603
Sex	100% female	7	404	0.066	-0.099	0.227		
	100% male	14	699	0.349	0.236	0.453	8.025	0.005

Abbreviation: k = number of samples, *n* = number of participants in moderator level, CI = confidence interval, LL = lower limit, UL = upper limit; # Groupings attempting to matching Craft et al. [11]; ^ Groupings attempting to match Woodman and Hardy [12].

**Table 5 ijerph-19-06381-t005:** Certainty of Results by Research Question.

Research Question		Certainty Basis		Rating
Q1: What was the overall relationship between a measure of state self-confidence and performance? Moreover, does the risk of individual study bias or across study bias (i.e., publication bias) moderate this relationship? No (not moderated by bias).		We replicated the overall confidence–performance relationship reported by Craft et al. [11] and Woodman and Hardy [12] with different inclusion criteria and many non-overlapping studies. Risk of individual study bias and publication bias had no impact on the overall relationship.		High

Q2: Did Terry’s [20] sport propositions moderate the confidence–performance relationship? Yes.		All compared moderator levels were in line with Terry’s [20] propositions. Individual vs. team findings were consistent with Craft et al. [11] and Woodman and Hardy [12] with many non-overlapping samples. Only inconsistency related to Craft et al. [11] large open vs. closed skill values (incongruent with Terry’s [20] propositions). Unable to replicate Craft et al. [11] given non-overlapping samples.		Moderate to High

Q3: Did the objectivity and reference of the performance measure moderate the confidence–performance relationship? Yes.		Significant difference between objective vs. subjective performance measure although both mean correlation values are small. Similar result with self-referenced vs. other-referenced performance measures.		Moderate
Q4: Did the time of self-confidence assessment prior to performance moderate the confidence–performance relationship? No.		The correlation values and 95% confidence intervals did not differ significantly by time of self-confidence assessment. Failed to replicate Craft et al. [11].		Moderate

Q5: Did selected individual difference variables, namely sex and athlete sport level, moderate the confidence–performance relationship? Yes (sex question).		Our meta-regression results and mean difference values were significant and replicated the Woodman and Hardy [12] finding that sex moderates the confidence–performance relationship.		High
No (sport level, athlete level question).		Small correlation was consistent with Woodman and Hardy [12]. Inconsistent but still wide 95% confidence intervals with the high standard sport level. Consistent with Craft et al. as all our values were small. However, we did not replicate Craft et al. [11] European Club relationship.		Moderate

## Data Availability

All data are contained in the article tables.

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
