# Peer review of "Revisiting the Self-Confidence and Sport Performance Relationship: A Systematic Review with Meta-Analysis"

_ijerph, 2022, doi:10.3390/ijerph19116381_

Round 1

Reviewer 1 Report

The introduction and background are very well written, and while being long, does provide a good overview and rationale for the need of the current study. There are a few comments below that could help to reduce the feeling of heaviness while reading such a long document.

General comments:

In the methods section, the writing seems to get a bit unusual. Statements like ‘This process took many meetings’ line 210, ‘they met frequently’ line 219, ‘after much attention to…’ line 221 adds nothing to the scientific process and should be removed. It appears as if the authors are trying to remind the reader of the complexity and time-burden of performing a SR which is unnecessary. All information that does not contribute to the ability to reproduce the study should be removed.

The document is long and takes some effort to read (as it the case in many SRs), but readability can be improved by reducing repetition between the writing and tables for example section 3.1: The PRISMA diagram contains the selection process so should not be repeated in text other than a brief summary and refer to table

The term ‘gender’ is used repeatedly when you appear to be referring to biological sex rather than gender orientation… please correct throughout to sex.

Specific comments:

Abstract:

There is only 1 sentence of intro (line 18) and then it moves directly onto the methods. No objectives/aims are presented in the abstract. The conclusion sentence (lines 31-33) is very confusing, please clarify this to align better with the findings of the study.

Materials and methods:

Line 169-171: ‘Inclusion criterion (d)….’ this sentence is not needed as, by definition, if they are not engaged in a competitive sport event or contest, then they do not meet the criteria are would be excluded.

Line 179: Google translate is not a valid method of translating research documents, there are often many errors. If research in other languages is to be used, then there should be a forward and backward translation performed by translators. How many papers were affected by this? How relevant is it to the outcomes? Also… if the studies were done in different languages then is there evidence of the outcome measures having been validated in those languages? Please clarify.

Lines 188 onwards:

The specific key words plus synonyms are not clearly indicated… this search could not be reproduced by the reader. Please improve this section. How were they combined in the search, use of Boolean operators, etc.

PRISMA diagram:

There are no reasons for exclusion of any of the articles. In the initial line, why would articles be excluded before screening? How can this happen?

The ‘Records screened’ block… I assume this is title/abstract screening?

‘Records sought for retrieval’: so these are the full texts, correct? What are the reasons for exclusion in this step?

Again, ‘records assessed for eligibility’ How is this different from the step above? Reasons for exclusion here?

Line 234: You state the Hoy et al risk of bias tool ‘guided’ your questions. If you changed this tool then you need to provide the evidence of the validation done before using an adapted tool.

Table 1: You have added the categories for the risk of bias in the text, but in the table just cite the number. This means the reader has to refer back into the text each time they want to work out which bias is being assessed. I suggest changing the the first column with numbers into the headings e.g. ‘target population’, ‘random selection’, etc.

For Bias #1: Would the high risk definition not actually be an exclusion criterion for the study?

Results:

See comment above re repetition of PRISMA diagram

Section 3.2, line 321: ‘Overall the studies contained several positive aspects…’ this is a very general statement that does not contribute to the understanding of the study, I suggest removing.

Discussion:

Line 420-425: These sentences seem to be direct contradictions of one another regarding time of play. You need to explain better how they are not contradicting each other or reword them to be clearer.

Section 4.1: line 486-488

If the title/abstract don’t clearly mention that they include something specific, then that study should be rated as a ‘maybe’ (i.e. not excluded at this level of screening), and the full text should be screened. This is potentially a major methodological flaw at a screening level that could have resulted in articles being excluded that actually were appropriate. Please double check your methods here and ensure that no eligible articles were excluded because of lazy screening processes!

Line 512: Study not registered, which is fine if not a requirement for the journal, but PROSPERO should be checked for duplication of the SR. Suggest checking and updating here as appropriate.

Author Response

Please see attached response.

Reviewer 2 Report

Title: Revisiting the Self-Confidence and Sports Performance Relationship: A Systematic Review with Meta-Analysis

Article Type: Systematic review and meta-analysis

Summary

The authors reviewed the state self-confidence and sports performance literature and examine the relationship between them and other existing and related variables using the meta-analytic technique. The authors searched some selected databases within the EBSCOhost platform and found about 41 related articles published between 1986 and 2020 from 15 countries and 24 sports. The results indicated that the confidence to athletic performance relationship is significantly different from zero, small in magnitude, and moderated by some sport, performance, and gender characteristics.

Minor suggestion:

This is suggested to add the name of the selected databases within the EBSCOhost such as SportDiscus, Psycinfo, … within the Abstract.

Given that in this manuscript, the state self-confidence was evaluated, why the authors didn't consider the level of competition and skill of the athletes? please speak about that in the manuscript.

Why did the authors add the non-English language articles to the manuscript? Did these articles have an English abstract? please explain that in the manuscript.

The authors have searched the EBSCOhost platform. why they didn't search other databases such as Scopus?

The image quality (Figure 1) is low, please increase the image quality.

Author Response

Please see attached response.

Reviewer 3 Report

I had the pleasure of reviewing the article entitled "Revisiting the Self-Confidence and Sport Performance Relationship: A Systematic Review with Meta-Analysis". 
This is a remarkable piece of research, both in terms of content and form. 
The authors strictly follow the PRISMA protocol. The research methodology is rigorous, all the elements are present in the text, and the complementary elements are presented in the appendices. 

The results are presented correctly and the discussion is adequate.

Thus, we have no contraindication to the publication of this research as it stands.

Author Response

Please see attached response.

Round 2

Reviewer 1 Report

Thank you for addressing the comments on the review. There are a number of times when you have responded to my comments, but not actually made the required changes in the document. I have pointed out my remaining concerns below:

Line 179: Google translate is not a valid method of translating research documents, there are often many errors. If research in other languages is to be used, then there should be a forward and backward translation performed by translators. How many papers were affected by this? How relevant is it to the outcomes? Also… if the studies were done in different languages then is there evidence of the outcome measures having been validated in those languages? Please clarify.

  • Response: Thank you for your comment and thoughts. We do not know of a reason Google Translate is not valid to find terms such as confidence, CSAI-2, correlations, participant information, etc. We found no points of confusion. If we would have, we would have sought out a native speaker as I (Marc Lochbaum) have with my past meta-analyses. The following two articles required Google Translate and included studies used a validated version of the CSAI-2 or CSAI-2R.

Pessoa da Costa, Y. et al., Desempenho técnico-tático e ansiedade competitiva no voleibol de praia com jovens atletas: efeito no resultado do jogo. / Technical-tactical performance and competitive anxiety in beach volleyball with young athletes: effect on the results of the match. Rev. Bras. Prescrição Fisio. Exer. 2019, 13(85), 876-885.

Pinto, M.F.; Vázquez, N. Ansiedad estado competitiva y estrategias de afrontamiento: su relación con el rendimiento en una muestra argentina de jugadores amateurs de golf. Competitive state anxiety and coping strategies: their relationship with the perfor-mance of an argentinean sample of amateur golf players. Rev. Psicol. Dep. 2013, 22(1), 47-52. 

Validation of the Brazilian version of the CSAI-2 was reported in:

Coelho EM, Vasconcelos-Raposo J, Mahl AC. Confirmatory factorial analysis of the Brazilian version of the Competitive State Anxiety Inventory-2 (CSAI-2). Span J Psychol. 2010 May;13(1):453-60. doi: 10.1017/s1138741600004005.

We can agree to disagree on the validity of Google Translate for scientific papers if the journal editors have no problem with its use.

Lines 188 onwards:

The specific key words plus synonyms are not clearly indicated… this search could not be reproduced by the reader. Please improve this section. How were they combined in the search, use of Boolean operators, etc.

My recommendation remains to place the final search strategy in the text as combined using Boolean operators, which will allow readers to easily reproduce this. Supplementary files are often not opened and having the executive summary in text makes it easier for the reader to understand and reproduce the search, rather than having to identify which is considered ‘Box 1’ or ‘Box 2’ for example.

PRISMA diagram:

‘Records sought for retrieval’: so these are the full texts, correct? What are the reasons for exclusion in this step? Again, ‘records assessed for eligibility’ How is this different from the step above? Reasons for exclusion here?

  • Response: Yes, full texts.
  • Response: It is not that different. Some articles took more time to be sure confidence assessment was before performance. However, the articles either met or did not meet the criteria. We eliminated the unnecessary boxes. Thank you for your questions. The process of stating all our hard work, and extra sentences, and repeated thoughts and boxes are gone.

It is usual to report the reasons for exclusions in these boxes (for example, incorrect outcome measure n=3). This again helps with reproducibility and it creates questions as to why the authors are unwilling to provide this information.

Line 234: You state the Hoy et al risk of bias tool ‘guided’ your questions. If you changed this tool then you need to provide the evidence of the validation done before using an adapted tool.

  • Response: Thank you for your thought here. The Hoy et al. did guide our questions. No one tool to our knowledge fits everyone. It could be for randomized control trials that tools exist with 100% fit. Per item 11 in the PRISMA (2020) guide, we followed the guide. The readers know what we did and can judge for themselves. The validation is as a team we worked on the categories and ratings within each.

Please add the information on the validation above to the paper.

For Bias #1: Would the high risk definition not actually be an exclusion criterion for the study?

  • Response: Thank for your point and detailed reading of our manuscript. We believe it is not a reason for exclusion. An example is the Laborde et al. publication. They used tennis players (not on a team like a college tennis team) and performed serves as in a match. We believe such studies fit our inclusion criteria – athletes and a sport performance.

Would ‘Sample includes recreational athletes….’ be a clearer definition for this than ‘Sample is a group, but not actually in the sport like an elite or college athlete’? As an example…

Section 4.1: line 486-488

If the title/abstract don’t clearly mention that they include something specific, then that study should be rated as a ‘maybe’ (i.e. not excluded at this level of screening), and the full text should be screened. This is potentially a major methodological flaw at a screening level that could have resulted in articles being excluded that actually were appropriate. Please double check your methods here and ensure that no eligible articles were excluded because of lazy screening processes!

  • Response: Thank you for your thoughts. We would not describe our search process as lazy. Confidence pulls up articles in the field of sport science (because of the search term sport) with confidence Those articles via the title and abstract are ones not requiring screening as the EBSCO framework bolds the searched term or terms. I (Marc Lochbaum) sat with my team member (MS) at screening time points going one by one with each article pulled up in the EBSCO search.

Again, why were studies excluded at title/abstract level for not mentioning confidence. If there is any uncertainty/possibility that the study could include relevant outcomes (even if not measured in abstract/title) then they should be screened during full-text screening. This needs to be explained more clearly rather than just included as a limitation.

Line 512: Study not registered, which is fine if not a requirement for the journal, but PROSPERO should be checked for duplication of the SR. Suggest checking and updating here as appropriate.

  • Response: Thank you for your suggestion. I (Marc Lochbaum) have written many meta-analyses. I did check PROSPERO. We did not write this sentence. Here is a renewed example search (completed 30 April 2022). When we began and up to this moment, I still do not see a duplicate.

While I appreciate your confirming your experience in MA, the point is that you have not mentioned this in your paper. Please include a brief statement in the paper to confirm that PROSPERO was searched before the review was performed to avoid duplication.

Author Response

Thank you for your thoughts. We addressed your comment in the attached document.
